# The Determination of Snow Albedo from Satellite Measurements Using Fast Atmospheric Correction Technique

**Alexander Kokhanovsky** [1,*] **, Jason E. Box** [2] **, Baptiste Vandecrux** [2] **, Kenneth D. Mankoff** [2] **, Maxim Lamare** [3] **, Alexander Smirnov** [4,5] **and Michael Kern** [6]

1   VITROCISET, Bratustrasse 7, D-64293 Darmstadt, Germany
2   Geological Survey of Denmark and Greenland (GEUS), 1050K Copenhagen, Denmark; jeb@geus.dk (J.E.B.); bav@geus.dk (B.V.); kdm@geus.dk (K.D.M.)
3   Météo-France, CNRS, CNRM, Centre d' Etudes de la Neige, 38400 Grenoble, France; maxim.lamare@meteo.fr
4   Science Systems and Applications, Inc., Lanham, MD 20706, USA; alexander.smirnov-1@nasa.gov
5   Biospheric Sciences Laboratory, NASA GSFC, Greenbelt, MD 20771, USA
6   Applications and Climate Department (EOP-SME), ESTEC/ESA, Science, 2200 AG Noordwijk, The Netherlands; Michael.Kern@esa.int
*   Correspondence: a.kokhanovsky@vitrocisetbelgium.com

**Abstract:** We present a simplified atmospheric correction algorithm for snow/ice albedo retrievals using single view satellite measurements. The validation of the technique is performed using Ocean and Land Colour Instrument (OLCI) on board Copernicus Sentinel-3 satellite and ground spectral or broadband albedo measurements from locations on the Greenland ice sheet and in the French Alps. Through comparison with independent ground observations, the technique is shown to perform accurately in a range of conditions from a 2100 m elevation mid-latitude location in the French Alps to a network of 15 locations across a 2390 m elevation range in seven regions across the Greenland ice sheet. Retrieved broadband albedo is accurate within 5% over a wide (0.5) broadband albedo range of the (N = 4155) Greenland observations and with no apparent bias.

**Keywords:** snow characteristics; optical remote sensing; snow albedo; PROMICE; Sentinel 3; OLCI; atmospheric correction; arctic aerosol

---

## 1. Introduction

There is a decreasing trend in both the extent and the reflective power of the terrestrial cryosphere with important climate change feedbacks [1–4]. The solar light reflectance from snow and ice has a bi-directional character, depending on the direction of illumination and on the observation direction, which can be measured using ground, airborne, and satellite optical instruments. Climate models utilize snow spectral plane albedo, which provides total reflected solar light power for a given wavelength and a given solar incidence angle that depends on location and time. Satellite measurements, of particular importance for studies of polar environment [4], are usually performed with a fixed observation geometry. Therefore, special procedures are needed to convert satellite-measured reflectance to a plane albedo [5,6]. The broadband plane albedo (BBA) can be derived using various parameterizations or by integration of the spectral plane albedo with account for the spectral snow irradiance at the snow surface [7]. The optical signals measured by satellite are influenced not just by light reflected from the surface but by atmospheric extinction, scattering and absorption. Therefore, atmospheric effects need to be removed to obtain accurate surface retrievals. Similarly, remote sensing of the atmosphere requires removal of the surface contribution to the observed signal. Atmospheric remote sensing

is more readily made in cases of dark underlying surfaces, such as the ocean. In polar regions, the retrievals of atmospheric aerosol load over bright snow and ice surfaces are challenging and often hardly possible because the signal is dominated by the bright surface and not by atmospheric aerosol.

A key task of this study is to provide an accurate determination of spectral and broadband plane albedo of snow and ice using satellite observations amid the challenge of atmospheric absorption by ozone, molecular light scattering and light scattering, and absorption by atmospheric aerosols. It is assumed that aerosol optical properties are known a priori, i.e., from aerosol climatology, forecasts, or ground measurements for the case of polluted snow/atmosphere. In the case of clean snow and atmosphere, we do not rely on any *a priori* information on atmospheric aerosol loading and properties in our snow and ice albedo retrieval technique. In any case, the generally low polar aerosol loading [4] reduces the influence of the aerosol contribution to the retrieved surface albedo. In the case of polluted snow, retrievals are performed outside strong atmospheric absorption bands (e.g., $O_2$ and $H_2O$). The ozone absorption effects are fully accounted for in the retrieval framework. While the algorithm is easily portable to other multi-spectral instruments observing the cryosphere from space, we present an application to data from the Ocean and Land Colour Instrument (OLCI) on board the European Union Copernicus Sentinel-3A satellite. The theoretical modeling of spectral snow reflectance is performed as in [6]. The earlier atmospheric correction used in [6], which appears in OLCI Snow Properties module incorporated in the European Space Agency (ESA) SeNtinel Application Platform (SNAP), can be biased in case of strong atmospheric pollution episodes (arctic haze, etc.) because it neglects scattering and absorption by liquid and solid particles suspended in atmosphere. This shortcoming of the previous algorithm as presented in [6] is eliminated in this study.

## 2. Materials and Methods

### 2.1. Theory

We perform the retrievals using separate retrieval chains for clean snow (Case 1 snow) and for polluted snow (Case 2 snow). Here, we use the analogy with the classification of Case 1 and Case 2 water as proposed in [8] (see also [9]). Case 1 water corresponds to relatively clean water where most of the absorption is due to phytoplankton, and Case 2 water contains other impurities including mineral particles. In our application, we define Case 1 as the situation where snow properties are determined just by snow grains without significant interference from impurities or living matter (cells, algae, etc.). The snow Case 1 is often met in Antarctica—far from any significant aerosol sources and limited algal populations. The areal extent of the clean dry snow areas on Greenland and Antarctica ice sheets makes the Case 1 snow dominant on a global scale. Additionally, a simplified atmospheric correction is possible in this case [6]. The selection of clean snow pixels is performed as follows. First, we check the reflectance in OLCI band 1. If it is larger than the dynamic threshold value (THV), it is assumed that the ground scene is covered by unpolluted snow (the majority of pixels in the terrestrial cryosphere). The THV is derived from the synthetic radiative transfer calculations for the assumed (default: 0.1) aerosol optical thickness at 550 nm (see Appendix A).

Case 1 snow

The simplified atmospheric correction for Case 1 snow is described in [6] and summarized below. It is based on the fact that the pure snow spherical albedo can be accurately parameterized using the following equation:

$$r_s = \exp\left(-\sqrt{\alpha(\lambda)l}\right), \tag{1}$$

where $\alpha(\lambda) = 4\pi\chi/\lambda$ is the bulk ice absorption coefficient for a given wavelength $\lambda$, $\chi$ (see e.g., https://atmos.washington.edu/ice_optical_constants/, last access: 07/01/2020) is the imaginary part of ice refractive index, and $l$ is the effective absorption length. The snow spectral reflectance $R_s$ is related to the snow spherical albedo, which is a three dimensional integral of $R_s$ with respect to solar

and viewing zenith angles and relative azimuthal angle (RAA) [6] via the following approximate equation [6]:

$$R_s = R_0 r_s^x,\tag{2}$$

where $x$ is a geometrical correction coefficient depending on $R_0$ and on the angular function $u$ [6] evaluated at the cosine of the solar zenith angle (SZA) $\mu_0$ or at the cosine of the viewing zenith angle (VZA) $\mu$:

$$x = \frac{u(\mu_0)u(\mu)}{R_0}\tag{3}$$

and we use the following approximation for the angular function [6]:

$$u(z) = \frac{3}{7}(1 + 2z).\tag{4}$$

The value of $R_0$ gives the non-absorbing underlying surface reflectance ($r_s = 1$). One can use OLCI measurements at 865 and 1020 nm to determine both effective absorption length and $R_0$ from Equation (2) under the assumption that the atmosphere does not affect the satellite signal at these channels [6].

The determined value of effective absorption length makes it possible to derive the snow spherical albedo at any wavelength using Equation (1). The plane albedo $r_p$ is defined via the integral of the azimuthally averaged reflection function with respect to the viewing zenith angle [6]. As a matter of fact, $r_p$ can be also derived from the spherical albedo using the following simple approximation [6]:

$$r_p = r_s^{u(\mu_0)}\tag{5}$$

or with account for Equation (1):

$$r_p = \exp\left(-u(\mu_0)\sqrt{\alpha(\lambda)l}\right)\tag{6}$$

Also, one can derive the underlying snow spectral reflectance function using Equations (1) and (2). Therefore, the procedure for the determination of Case 1 spectral albedo from space is straightforward. It was validated in [6]. Generally, the errors in the retrieved albedo are below 1–3% depending on the wavelength $\lambda$.

Case 2 snow

The retrievals for the Case 2 snow are more complicated. In this case, the satellite measurements of snow spectral reflectance in the visible are influenced by various pollutants or living matter (cells, algae, etc.). Therefore, there is no way to estimate snow spectral reflectance/albedo in the visible using measurements in the near infrared as it is done for the Case 1 snow (see above). Then, we use yet another approach described below.

The top-of-atmosphere reflectance for the atmosphere-underlying snow system can be presented in the following way [10,11]:

$$R_{meas} = R_{ag} + \frac{T_{ag}r_s}{1 - r_{ag}r_s},\tag{7}$$

where $R_{ag}$ is the atmospheric contribution to the measured signal, $r_{ag}$ is the spherical albedo of the atmosphere, $r_s$ is the bottom-of-atmosphere snow spherical albedo, and $T_{ag}$ is atmospheric transmittance from the top-of-atmosphere to the underlying surface and back to the satellite position. In the case of Lambertian underlying surfaces, the underlying surface reflectance does not depend on solar and viewing observation directions, and Equation (7) is valid with $r_s = R_s$, where $R_s$ is the underlying Lambertian surface reflectance. The snow is not exactly the Lambertian reflector; therefore, we replace $r_s$ in the numerator of Equation (7) by the snow reflectance [see Equation (2)]. Such an approximation makes it possible to have the correct limit for the top-of-atmosphere reflectance [see Equation (2)] in the case of absence of atmosphere. The term in the dominator of Equation (7) accounts for multiple

reflections between snow and atmosphere, and the account for the snow reflectance directional nature in the dominator of this equation is of secondary importance. Then, it follows:

$$R_{meas} = R_{ag} + \frac{T_{ag}R_0 r_s^x}{1 - r_{ag}r_s}. \tag{8}$$

The reflectance of non-absorbing snow $R_0$ in Equation (8) is calculated using simple analytical approximation, as discussed in [6]. We do not derive the value of $R_0$ from OLCI measurements themselves because such a derivation for the polluted snow can be influenced by the type and the load of pollutants.

We use channels that are not influenced by water vapor and oxygen absorption effects, although we account for the ozone absorption effects. Equation (8) is very general and valid outside and inside molecular absorption bands. We account for the ozone absorption in a simplified way. Namely, we derive free of ozone absorption top-of-atmosphere reflectance $R_c$ using the following equation: $R_c = \frac{R_{meas}}{T_{O3}}$, where $T_{O3}$ is the atmospheric transmittance with account for the ozone absorption (see Appendix A). Then, Equation (8) is transformed to a simplified approximation:

$$R_c = R_a + \frac{T_a R_0 r_s^x}{1 - r_a r_s}, \tag{9}$$

where the functions $R_a$, $r_a$, $T_a$ (see Appendix A) have the same meaning as $R_{ag}$ $r_{ag}$, $T_{ag}$, respectively, except for atmosphere not influenced by gaseous absorption processes (e.g., ozone absorption). The spherical albedo of underlying snow surface can be found from Equation (9) provided that the aerosol model is known. In this case, the snow spherical albedo $r_s$ is the only unknown parameter in Equation (9) and can be readily calculated, solving the transcendent Equation (9) with respect to $r_s$. For the wavelengths where the aerosol contribution is low and can be neglected, $R_a \sim 0$, $r_a \sim 0$, $T_a \sim 1$, and an analytical solution of Equation (9) is possible:

$$r_s = \left(\frac{R_c}{R_0}\right)^{1/x}, \tag{10}$$

where the analytical expression for $R_0$ is given in [6]. The functions $R_a$, $T$, and $r_a$ depend on aerosol and molecular scattering parameters and can be stored in look-up-tables for various aerosol models. Because aerosol load is weak in the Arctic and Antarctica, various approximations for the functions mentioned above can be used. In particular, we calculate these functions in the framework of approximations described in the Appendix A. We solve the transcendent Equation (9) with respect to $r_s$ for all OLCI wavelengths free of water vapor and oxygen absorption in the Case 2 snow.

The broadband albedo (BBA), either plane or spherical, is calculated from the spectral plane or the spherical albedo using the integration between the wavelengths $\lambda_a$ and $\lambda_b$ as shown below [7]:

$$\bar{r}_{p,s}(\lambda_1, \lambda_2) = \frac{\int_{\lambda_a}^{\lambda_b} r_{p,s}(\lambda)F(\lambda)d\lambda}{\int_{\lambda_a}^{\lambda_b} F(\lambda)d\lambda}. \tag{11}$$

where $F(\lambda)$ is the incident solar flux at the snow surface, and $r_{p,s}(\lambda)$ is plane ($p$) or spherical ($s$) albedo depending on whether plane or spherical BBA $\bar{r}_{p,s}(\lambda_a, \lambda_b)$ is to be calculated. The indices $a$ and $b$ signify the wavelengths $\lambda$ used. We assume that the incident solar flux can be approximated by the following analytical function:

$$F(\lambda) = f_0 + f_1 \exp(-\psi\lambda) + f_2 \exp(-\gamma\lambda), \tag{12}$$

where we ignore rapid oscillations of $F(\lambda)$, which are due to gaseous absorbers. This is possible because $r_{p,s}(\lambda)$ is a continuous function, which acts as a filter of high frequencies. The coefficients in Equation (12) are derived from the fit of $F(\lambda)$ calculated using the Santa Barbara DISORT Radiave

Transfer (SBDART) code [12] to Equation (12) in the spectral range 0.3–2.4 μm. They are given in Table 1. The calculations of $F(\lambda)$ are performed at the parameters listed in Table 2 for the rural aerosol model [13]. Clearly, $F(\lambda)$ depends on the location and the time. We find that the choice of aerosol model in the calculation of $F(\lambda)$ only weakly influences the calculations of BBA (see Equation (11)). The spectral snow albedo needed as input for SBDART is calculated assuming clean snow with the effective diameter of spherical ice grains equal to 0.25 mm. Generally, the results are only weakly sensitive to the variation of the function $F(\lambda)$ [7]. We therefore assume solar flux independent from the location of the retrieval and from solar zenith angles.

**Table 1.** The coefficients of approximation given by Equation (12).

| $f_0$ | $f_1$ | $f_2$ | $\psi$, 1/Microns | $\gamma$, 1/Microns |
|---|---|---|---|---|
| $3.238 \times 10^1$ | $-1.6014033 \times 10^5$ | $7.95953 \times 10^3$ | $1.778 \times 10^3$ | $2.489 \times 10^1$ |

**Table 2.** The parameters of calculations performed using the Santa Barbara DISORT Radiative Transfer (SBDART).

| Parameter | Value |
|---|---|
| Water vapor column | $2.085$ g/m$^2$ |
| Total ozone column | 350 Dobson Units (DU) |
| Tropospheric ozone | 34.6 DU |
| Aerosol Optical Thickness (AOT) at 550 nm | 0.1 |
| Altitude | 825 m |
| Solar zenith angle | 60 degrees |

For the Case 1 snow, the broadband albedo is calculated numerically using Equations (1), (5), (11), and (12) in the spectral range 0.3–2.4 micrometers. Also, other limits of integration can be used (say, to derive visible or near-infrared BBA).

For the Case 2 snow, the spherical albedo is known only for selected OLCI channels as derived from Equation (9). Therefore, we use interpolation to get the spherical albedo between the measurement points needed for the evaluation of integral (11). For the spectral range below 865 nm, we use:

$$r_s = c\lambda^2 + b\lambda + a. \tag{13}$$

While, for wavelengths larger than 865 nm, we use:

$$r_s = \sigma \exp(-\epsilon\lambda). \tag{14}$$

We use the dependencies as shown in Equations (13) and (14) because we find that the measurements can be approximated by the second order polynomial for the spectral range below 865 nm and the exponential function for the wavelengths above 865 nm. The coefficients $(a, b, c)$ are found separately for the intervals 400–709 nm and 709–865 nm using the following wavelength triplets: (400, 560, 709 nm) and (709, 753, and 865 nm), respectively.

The coefficients $(\epsilon, \sigma)$ are derived from OLCI measurements at 865 and 1020 nm at the values of $R_{meas}$ (1020 nm) equal to or smaller than 0.5. Otherwise, Equation (1) [and not Equation (14)] is used at $\lambda > 865$ nm with the effective absorption length derived from the value of spherical albedo at 1020 nm. We use different approaches for the pixels with small and large values of $R_{meas}$ (1020 nm) because the case of comparatively large values of $R_{meas}$ (1020 nm) corresponds to snow. Otherwise, ice or extremely dirty snow is present. Then, Equation (1) is not valid.

Integral (11) for the spherical broadband albedo with account for Equations (12)–(14) can be evaluated analytically. The answer is:

$$\bar{r}_s(\lambda_a, \lambda_b) = \bar{r}_{s1}(\lambda_a, \lambda_1) + \bar{r}_{s2}(\lambda_1, \lambda_2) + \bar{r}_{sd}(\lambda_2, \lambda_b), \tag{15}$$

where

$$\bar{r}_{sj}(\lambda_a, \lambda_b) = a_j + (b_j K(\lambda_a, \lambda_b) + c_j L(\lambda_a, \lambda_b))/J(\lambda_a, \lambda_b),$$

$$\bar{r}_{sd}(\lambda_a, \lambda_b) = M(\lambda_a, \lambda_b)/J(\lambda_a, \lambda_b),$$

$$J(\lambda_a, \lambda_b) = f_0 j_0 + f_1 i_1(\psi) + f_2 i_1(\gamma),$$

$$K(\lambda_a, \lambda_b) = f_0 k_0 + f_1 i_2(\psi) + f_2 i_2(\gamma), \qquad (16)$$

$$L(\lambda_a, \lambda_b) = f_0 l_0 + f_1 i_3(\psi) + f_2 i_3(\gamma),$$

$$M(\lambda_a, \lambda_b) = \sigma(f_0 i_0(\epsilon) + f_1 i_0(\epsilon + \psi) + f_2 i_0(\epsilon + \gamma)).$$

Here, the coefficients $a_j$, $b_j$, $c_j$ are the same as presented in Equation (13) with $j = 1$ for the first spectral interval (0.3–0.709 microns) and $j = 2$ for the second spectral interval (0.709–0.865 microns). $\lambda_a = 0.3 \ \mu m, \lambda_1 = 0.709 \ \mu m, \lambda_2 = 0.865 \ \mu m, \lambda_b = 2.4 \ \mu m, J(\lambda_a, \lambda_b)$ is the integral given in the dominator in Equation (11) [evaluated analytically with account for Equation (12)] and:

$$j_0 = \lambda_b - \lambda_a, \ k_0 = (\lambda_b^2 - \lambda_a^2)/2, \ l_0 = (\lambda_b^3 - \lambda_a^3)/3,$$

$$i_1(v) = (\exp(-v\lambda_a) - \exp(-v\lambda_b))/v,$$

$$i_2(v) = \left( \frac{1}{v^2} + \frac{\lambda_a}{v} \right) \exp(-x\lambda_a) - \left( \frac{1}{v^2} + \frac{\lambda_2}{v} \right) \exp(-v\lambda_b) \qquad (17)$$

$$i_3(v) = \left( \frac{2}{v^3} + \frac{2\lambda_a}{v^2} + \frac{\lambda_a^2}{v} \right) \exp(-v\lambda_a) - \left( \frac{2}{v^3} + \frac{2\lambda_b}{v^2} + \frac{\lambda_b^2}{v} \right) \exp(-v\lambda_b).$$

At the $R$ (1020 nm) equal to or above 0.5, the analytical expression for the BBA cannot be derived (because one accounts for Equation (1) and not Equation (13) in Equation (11)). Then, the numerical integration procedure is followed.

The broadband plane albedo is calculated in a similar way as a broadband spherical albedo using Equation (5) for the transformation of spherical to plane albedo.

This concludes the description of this new fast radiative transfer Snow and ICE surface albedo retrieval (SICE) that accounts for atmospheric scattering and absorption effects. The SICE algorithm can be considered as an update of the previous version of the algorithm (called S3Snow [6]) that appeared in the Snow Properties module of SNAP.

*2.2. Validation*

2.2.1. Snow Spectral Albedo

The validation of spectral albedo for the case of clean snow is reported in [6], where a detailed description of ground and satellite measurements, not repeated here, may be found. In the case of polluted snow, we follow a different procedure than that suggested in [6]. Here, we use the improved atmospheric correction, which explicitly accounts for molecular/aerosol light scattering and absorption effects. The results for the French Alps, Col du Lautaret validation site (45.041288N, 6.410557E, 2100 m a.s.l.) on 17 April, 2018 (Figure 1a) confirm that the current SICE planar albedo retrieval has a higher accuracy as compared to the earlier S3Snow algorithm [6] for the cases studied. Also, unlike the S3Snow retrievals, there is a possibility to vary aerosol load in the framework of the updated retrieval, which is currently not the case for S3Snow plane albedo retrieval results. As it follows from Figure 1b, the variation of aerosol optical thickness (AOT) (500 nm) in the range 0.07–0.35 does not change the plane albedo retrieval accuracy considerably (above 3% for the case studied). Note that the AOT (500 nm) obtained from the Copernicus Atmospheric Monitoring Service near-real-time forecast product (https://www.ecmwf.int/en/about/what-we-do/environmental-services/copernicus-atmosphere-monitoring-service, last access: 07/01/2020) is 0.125 for the case studied. We conclude that

the precise information on the spectral aerosol optical thickness is not needed for the accurate snow spectral albedo retrievals for the low aerosol load characteristic for the Col du Lautaret validation site located at 2,100 m a.s.l. [4]. It should be pointed out that the discrepancy of satellite and ground plane albedo measurements is not solely due to the retrieval but also partially due to surface spatial inhomogeneity (local scale effects vs. much broader scale satellite pixel effects), time difference between satellite and ground measurements, and influence of 3-D effects from surrounding mountains.

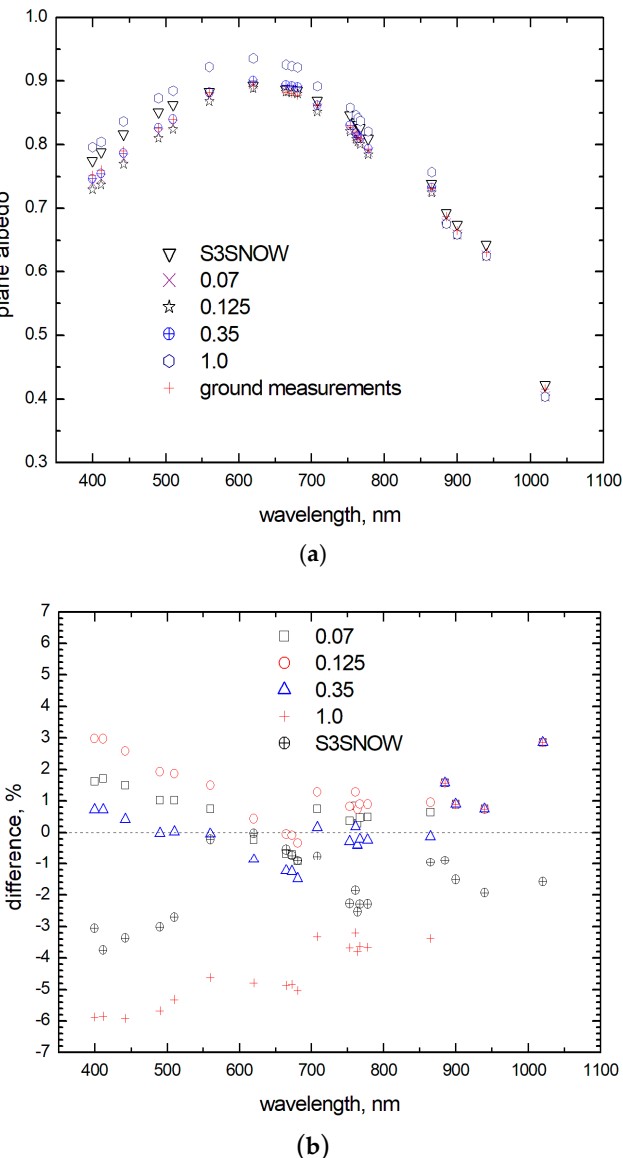

**Figure 1.** (**a**) The plane albedo of dust laden snow retrieved from satellite measurements and measured on ground on 17 April 2018 at the at the Col du Lautaret validation site in the French Alps. In the retrieval process, four values of aerosol optical thickness (AOT) at 500 nm are applied (0.07, 0.125, 0.35, and 1.0). The retrievals using S3Snow are also shown. (**b**) The differences in spectral plane albedo of dust laden snow retrieved from satellite measurements and measured on ground on 17 April 2018 at the at the Col du Lautaret validation site in the French Alps. In the retrieval process, four values of AOT at 500 nm in the framework of Snow and ICE surface albedo retrieval (SICE) are assumed (0.07, 0.125, 0.35, and 1.0). The differences of satellite retrievals and ground measurements of plane albedo using S3Snow are also presented.

### 2.2.2. Snow Broadband Albedo

We validate broadband albedo measured in the 0.3–2.4 micron wavelength range using ground measurements from fifteen Programme for the Monitoring of the Greenland Ice Sheet (PROMICE) automatic weather stations, with albedo data already described with more detail in [6]. In comparisons presented here, the closest hourly observations are considered. Occasionally for northern sites (KPC_L, KPC_U), there are multiple comparisons each day. Clear sky conditions are estimated from downward longwave irradiance data after [6]. A total of 4,146 individual comparisons are made. The PROMICE BBA data include a correction for measurement platform obstruction of the radiometer field of view [14] that increases average PROMICE albedo by 0.034.

At the PROMICE SCO_U location (Figures 2 and 3, Table 3) on the eastern ice sheet, where clear sky conditions are common, we observe in three May–September years (2017, 2018, 2019) over a wide (0.52) BBA range (from 0.33 indicative of impurity rich bare ice to 0.85 indicative of dry clean snow cover) a very similar temporal pattern in both the ground observations and from the OLCI retrievals.

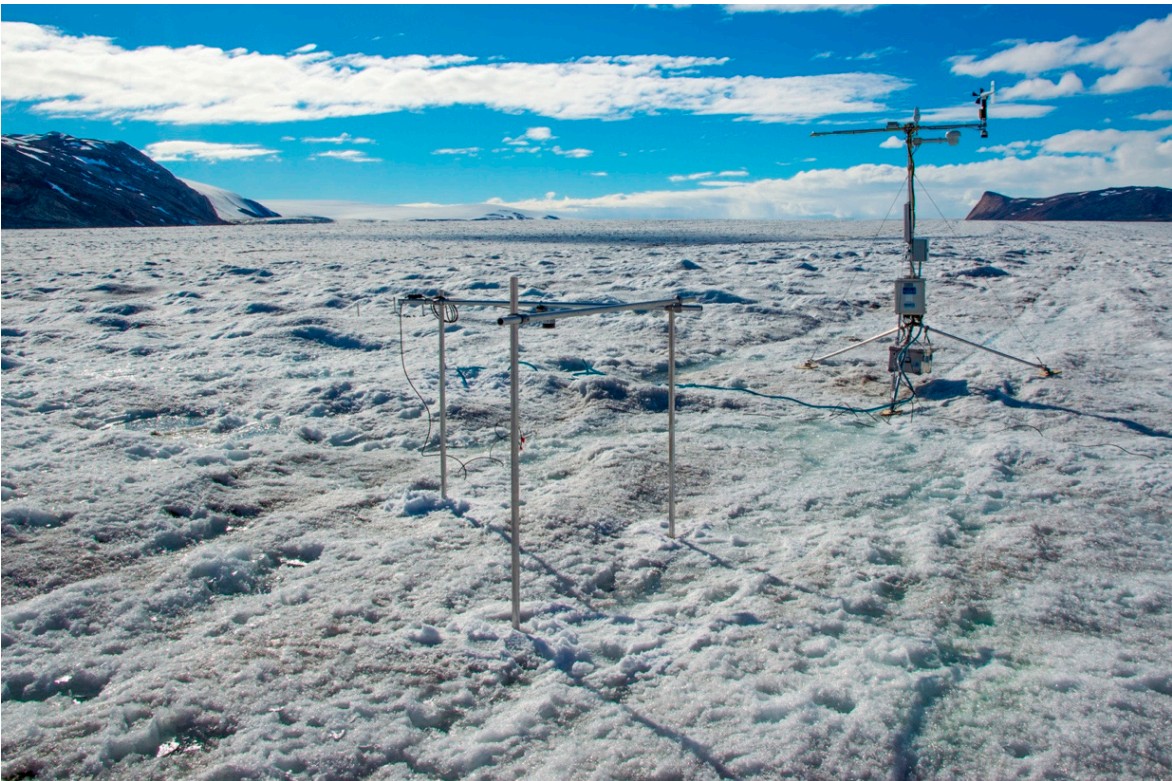

**Figure 2.** The SCO_U Programme for the Monitoring of the Greenland Ice Sheet (PROMICE) automatic weather station with radiometer on main station in the background. The photo is from 4 August 2017 under typical late ablation season surface conditions with a bare ice surface having relatively high mineral dust and biological absorbers.

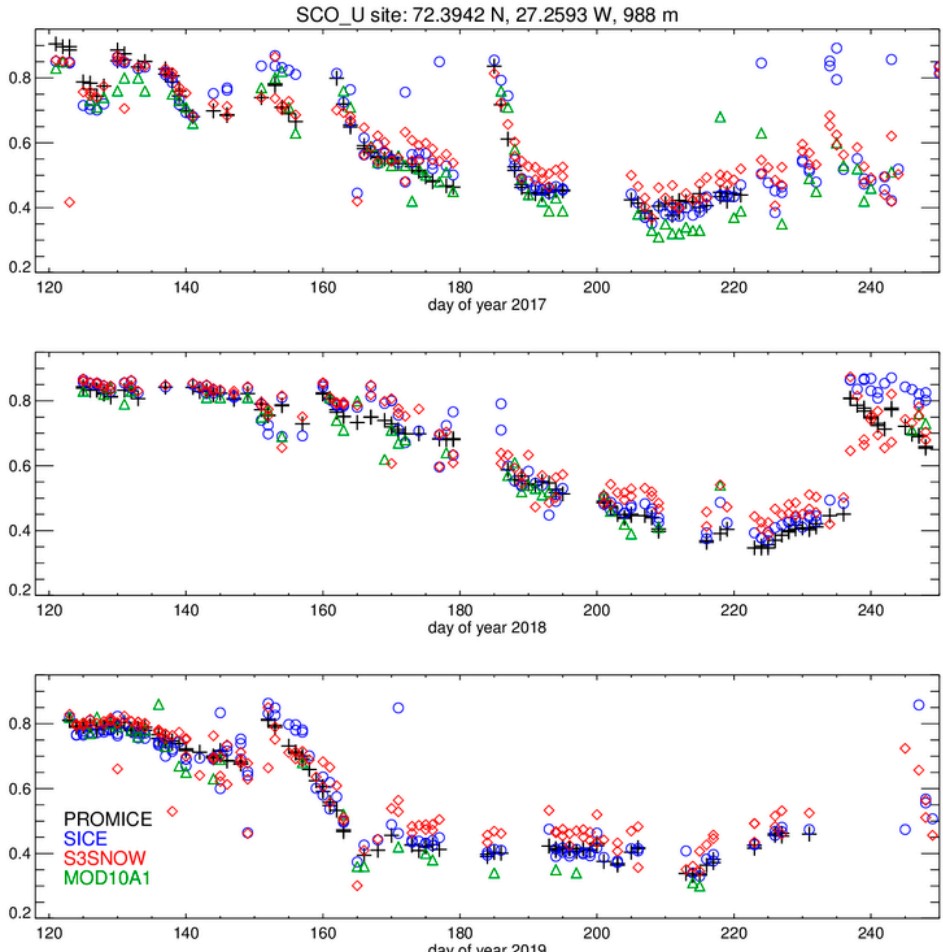

**Figure 3.** Example of the broadband albedo (BBA) time series from the SCO_U PROMICE automatic weather station site (Table 3) in comparison to the BBA retrievals from the current retrieval (SICE), the earlier Ocean and Land Colour Instrument (OLCI) retrieval (S3SNOW) after [6] and for the National Aeronautics and Space Administration (NASA) Moderate Resolution Imaging Spectroradiometer (MODIS) MOD10A1 product [15]. The symbols have the following meanings: PROMICE (crosses), SICE (circles), S3SNOW (rhombus), MOD10A1 (triangles). The several outliers for the SICE product (too high BBA) are related to the problem with our automatic cloud detection procedure.

Among the fifteen PROMICE locations spanning a wide spatial scale [2076 km north–south (18.9° latitude) and 2390 m in elevation], SICE BBA agreement is as high as is realistic to expect with unattended automatic weather station (AWS) observations even though we filter the sample to select cases when AWS tilt recordings are under 1 degree and downward longwave-derived cloud index is under 0.3 [6] (Table 3). It is very encouraging to find: regression slopes averaging insignificantly from unity; an average multi-site correlation coefficient of 0.869 and an average root-mean-square difference (RMSD) of 0.056. The relatively low correlation at the EGP site is more due to the relatively small (~0.1) seasonal fluctuation than a higher error at that site. In idealized circumstances, i.e., with tilt well under 1 degree and evidence of absolutely clear sky conditions from smooth downward shortwave diurnal curves, agreement on a case by case basis can be better than 0.02 (not shown), but such high agreement is not to be expected from a large sample of automatically-selected cases.

**Table 3.** Validation statistics for broadband albedo at fifteen PROMICE Greenland ice sheet ground stations. Here, N is the number of closest hourly observations.

| PROMICE Station Name | Latitude, Degrees North | Longitude, Degrees | Elevation, m Above Sea Level | Regression Slope | Regression Constant | Correlation Coefficient | Mean SICE BBA—PROMICE BBA | RMSD | N |
|---|---|---|---|---|---|---|---|---|---|
| KPC_L | 79.908 | −24.080 | 366 | 0.782 | 0.150 | 0.958 | −0.012 | 0.067 | 447 |
| KPC_U | 79.833 | −25.163 | 865 | 0.746 | 0.192 | 0.800 | 0.013 | 0.045 | 431 |
| SCO_L | 72.223 | −26.818 | 459 | 1.011 | 0.004 | 0.958 | −0.010 | 0.051 | 268 |
| SCO_U | 72.394 | −27.259 | 988 | 1.002 | 0.015 | 0.971 | −0.016 | 0.045 | 349 |
| QAS_L | 61.031 | −46.849 | 270 | 0.960 | 0.021 | 0.972 | −0.003 | 0.049 | 126 |
| QAS_U | 61.099 | −46.833 | 621 | 0.814 | 0.127 | 0.868 | −0.006 | 0.083 | 149 |
| QAS_M | 61.175 | −46.820 | 892 | 0.795 | 0.133 | 0.961 | −0.009 | 0.066 | 122 |
| NUK_L | 64.482 | −49.538 | 527 | 0.551 | 0.192 | 0.743 | −0.040 | 0.064 | 196 |
| NUK_U | 64.510 | −49.271 | 1119 | 0.922 | 0.040 | 0.810 | 0.013 | 0.090 | 190 |
| KAN_L | 67.095 | −49.953 | 664 | 0.944 | 0.029 | 0.863 | −0.004 | 0.028 | 194 |
| KAN_U | 67.000 | −47.027 | 1842 | 0.501 | 0.401 | 0.740 | 0.017 | 0.031 | 176 |
| UPE_L | 72.893 | −54.295 | 211 | 1.383 | −0.218 | 0.884 | −0.013 | 0.076 | 241 |
| UPE_U | 72.887 | −53.585 | 929 | 1.085 | −0.017 | 0.886 | −0.041 | 0.077 | 264 |
| THU_L | 76.400 | −68.266 | 566 | 1.013 | 0.010 | 0.970 | −0.018 | 0.048 | 346 |
| THU_U | 76.420 | −68.146 | 761 | 0.978 | −0.017 | 0.853 | 0.034 | 0.064 | 327 |
| EGP | 75.625 | −35.973 | 2660 | 0.550 | 0.372 | 0.659 | 0.009 | 0.014 | 320 |
| | | average | 859 | 0.877 | 0.090 | 0.869 | −0.005 | 0.056 | 259 |
| | | st.dev. | 620 | 0.226 | 0.153 | 0.097 | 0.020 | 0.021 | 103 |

RMSD: root-mean-square difference.

The current approach, advanced from that reported in [6] by the improved atmospheric correction, has increased agreement with the PROMICE data and an apparent accuracy that also exceeds that in the comparison with the National Aeronautics and Space Administration (NASA) Moderate Resolution Imaging Spectroradiometer (MODIS) MOD10A1 [15] albedo product (Figures 4 and 5). Examples are made for the southern Greenland ice sheet QAS_L PROMICE location and the northwestern ice sheet THU_L PROMICE location (Figure 4, right).

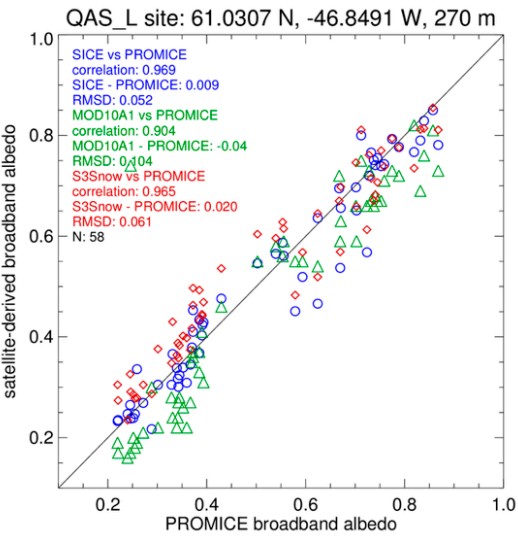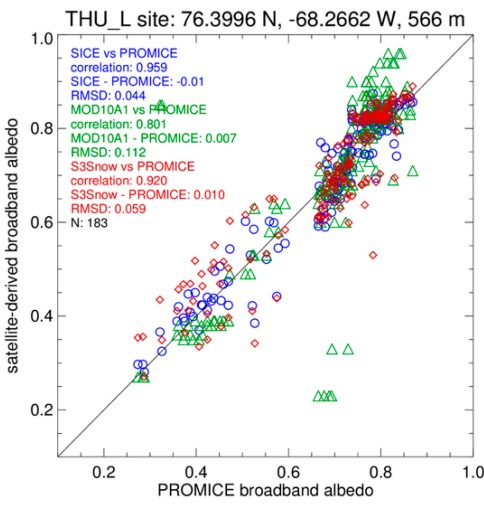

**Figure 4.** Examples of satellite derived and ground observations of snow and bare ice albedo. The left figure includes both the S3 Snow and the current processing results for the site QAS_L. The comparison uses cases only when all retrievals are available, i.e., including those from MODIS MOD10A1 [15]. The right figure is the same except for the site THU_L. The symbols have the following meanings: PROMICE (crosses), SICE (circles), S3SNOW (rhombus), MOD10A1 (triangles).

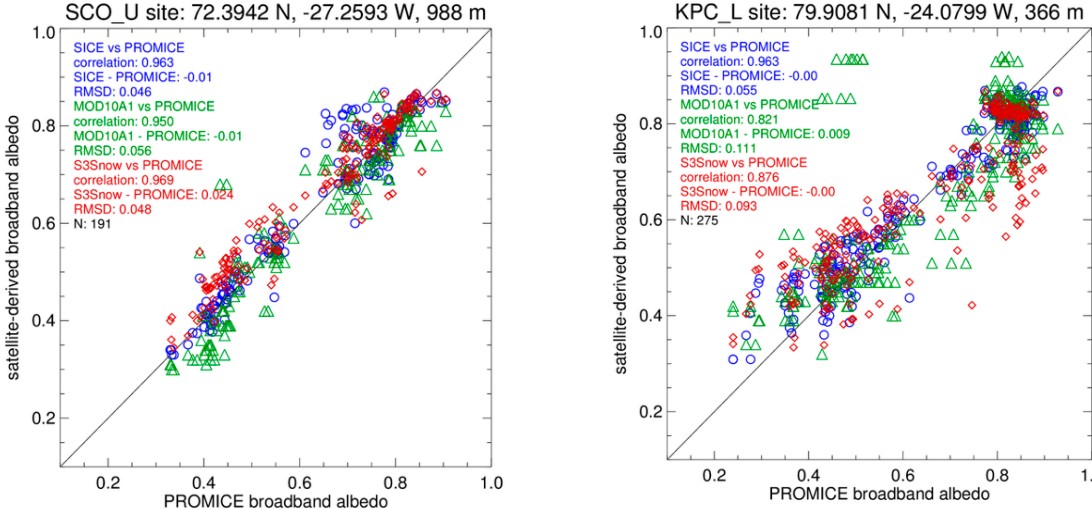

**Figure 5.** Same as Figure 4 but for the SCO_U location (also illustrated in Figure 3) (**left**) and for a northeastern location (KPC_L) (**right**).

## 3. Discussion

Standard underlying surface albedo retrieval algorithms based on single view observations can be corrected for surface anisotropy effects using multiple day observations of reflected solar light for a given site to cover necessary illumination/observation geometries needed for the respective integration procedures with respect to the corresponding zenith, viewing, and relative azimuth angles. In our approach, we use the analytical relationship between the bottom-of-atmosphere reflectance and the spherical albedo for clean snow underlying surface (Equations (1) and (2)) in the near infrared (865 and 1020 nm), where atmospheric contribution to the signal as registered on a satellite is small, to derive the snow spherical albedo from measurements at a fixed illumination/observation geometry. We underline that our major assumption is that atmospheric influences on OLCI measurements at 865 and 1020 nm in polar regions are weak and can be neglected. Then, top of atmosphere (TOA) reflectance almost coincides with bottom of atmosphere (BOA) reflectance (see Equation (2)). In the framework of our technique, the multiple day observations for the same site are not required, and the snow albedo for a given place can be derived in approximately one hour after the satellite acquisition time. In the case of polluted snow, the spherical albedo is found from Equation (9) for an assumed aerosol model. The technique also incorporates the calculation of plane spectral and broadband albedo. We find that the errors for the Case 1 snow are usually in the range 1–2% in the visible as compared to the ground measurements [6]. They can increase to 3–5% for the spectral albedo in the near IR and for polluted snow. The retrievals for the dark pixels are less accurate because the underlying theory is more accurate for the case of bright pixels [6]. The retrieval method presented here is the extension of the technique described in [6] for the cases where the aerosol load cannot be neglected.

Water exists in three thermodynamic phases (liquid, solid, gas) both in the atmosphere and in the underlying surface. The separation of clean (Case 1) and polluted (Case 2) waters has been useful in oceanic remote sensing using spaceborne observations. We show that a similar separation of satellite retrievals for clean and polluted snow areas (Case 1 and Case 2 snow) is useful in remote sensing of snow from space. Actually, a similar separation of cases is of importance in cloud remote sensing, where modern cloud remote sensing algorithms are based on the assumption of clean (Case 1) clouds. The polluted (Case 2) clouds exist, but up to now, their study is much less advanced.

In this paper, we propose fast snow albedo retrieval techniques for both Case 1 and Case 2 snow. The results for the clean snow are more accurate and robust. The retrievals for the Case 2 snow are less accurate and are based on the simplified atmospheric correction procedure specified in Equation (9) and the general relationship between reflectance and albedo given by Equation (10). We find that

the influence of the aerosol load on the retrieval of the snow surface albedo is weak in the case of small atmospheric aerosol optical thickness characteristic for Arctic and alpine areas. As a matter of fact, Equation (10) performs well not only for snow but also for other types of weakly absorbing and strongly scattering media, such as clean and polluted bare ice. Therefore, the technique proposed here can be used to study the albedo of terrestrial bare ice surfaces, as demonstrated in Figures 3–5, where low albedo values correspond not to snow but to ice underlying surfaces.

## 4. Conclusions

Through comparison with independent ground observations, the proposed fast atmospheric correction technique is shown to perform accurately in wide a range of conditions from a 2100 m elevation mid-latitude location in the French Alps to a Greenland ice sheet network of 15 locations spanning a 2076 km north–south, 18.9 degrees latitude, and 2390 m in elevation. It should be pointed out that snow albedo satellite retrievals are often biased due to the assumed shapes of ice grains (spheres, columns, fractal particles, etc.) used in the retrieval process. We use the notion of the effective absorption length in this work. It makes it possible to include all shape-dependent constants in the value of effective absorption length determined from the satellite measurements themselves. This reduces the snow grain shape effect on the retrievals (at least in the OLCI spectral range). The atmospheric correction is performed assuming the aerosol model and the aerosol optical thickness ahead of retrievals. The associated errors do not lead to considerable errors in the retrieved snow albedo in the case of low aerosol load, as demonstrated in Figure 1.

The current approach, advanced from that reported in [6] by the improved atmospheric correction, has increased agreement with ground observations and an apparent accuracy that also exceeds that of the NASA MODIS MOD10A1 [15] broadband albedo product.

A next step is to process the full OLCI catalogue from Sentinel-3A and B satellites over 100% snow or ice covered areas of our planet using this new algorithm. The product would offer the climate research community a new and enhanced quality snow and ice albedo product, which will lead to the advancement of our knowledge of snow albedo effects on the terrestrial climate change [1,2].

**Author Contributions:** Conceptualization, A.K., J.E.B.; methodology, J.E.B., A.K., A.S.; software, A.K., J.E.B., K.D.M.; validation, B.V., A.K., J.E.B., M.L.; formal analysis, A.K., A.S., J.E.B.; resources, J.E.B.; data curation, J.E.B., K.D.M., B.V.; writing—original draft preparation, A.K., J.E.B.; writing—review and editing, A.K., J.E.B., M.L., B.V., A.S.; visualization, A.K., J.E.B., supervision, M.K., J.E.B.; project administration, M.K., J.E.B.; funding acquisition, J.E.B., A.K. All authors have read and agreed to the published version of the manuscript.

**Funding:** Agency (ESA) studies; the Scientific Exploitation of Operational Missions (SEOM) Sentinel-3 Snow (Sentinel-3 for Science, Land Study 1: Snow), ESRIN contract 4000118926/16/I-NB and the ESRIN contract 4000125043-ESA/AO/1-9101/17/I-NB EO SCIENCE FOR SOCIETY. Additional support came from The Program for the Monitoring of the Greenland Ice Sheet (PROMICE), part of the Danish Energy Agency through the DANCEA program. M.L. was funded by BNP Paribas foundation, APR CNES MIOSOTIS, and EBONI ANR-16-CE01-006. CNRM/CEN is part of Labex OSUG@2020 (investissement d'avenir-ANR10 LABX56).

**Acknowledgments:** The authors thank Dumont, M. and Picard, G. for providing snow spectral albedo data used in this work for validation purposes and also Rozanov, V. for providing data given in Table A1. The authors are grateful to the AERONET staff, PIs and site managers.

**Conflicts of Interest:** The authors declare no conflict of interest.

## Appendix A. Atmospheric Radiative Transfer: Simple Approximations

The top of atmosphere reflectance $R_a$ for a clear atmosphere can be presented in the following way using the Sobolev approximation [16]:

$$R_a = R_{ss} + R_{ms}, \tag{A1}$$

where single scattering contribution:

$$R_{ss} = M(\tau)p(\theta) \tag{A2}$$

and multiple light scattering contribution is approximated as:

$$R_{ms} = 1 + M(\tau)q(\mu_0, \mu) - \frac{N(\tau)}{4 + 3(1-g)\tau}, \tag{A3}$$

where,

$$M(\tau) = \frac{1 - e^{-m\tau}}{4(\mu_0 + \mu)}, \quad N(\tau) = f(\mu_0)f(\mu) \tag{A4}$$

$$f(\mu) = 1 + \frac{3}{2}\mu + \left(1 - \frac{3}{2}\mu\right)e^{-\frac{\tau}{\mu}}, \quad m = \mu_0^{-1} + \mu^{-1}, \tag{A5}$$

$$q(\mu_0, \mu) = 3(1+g)\mu_0\mu - 2(\mu_0 + \mu). \tag{A6}$$

Here, $\mu_0$ is the cosine of the solar zenith angle (SZA), $\mu$ is the cosine of the viewing zenith angle (VZA), and $\theta$ is the scattering angle defined as:

$$\cos\theta = -\mu_0\mu + s_0 s \cos\varphi, \tag{A7}$$

$\varphi$ is the relative azimuthal angle (equal to 180 degrees minus OLCI relative azimuthal angle), $s_0$ is the sine of the SZA, $s$ is the sine of the VZA, $\tau$ is the atmospheric optical thickness, $p(\theta)$ is the phase function, and $g$ is the asymmetry parameter. It is determined by the following expression:

$$g = \frac{1}{2}\int_0^\pi p(\theta)\sin\theta\cos\theta d\theta. \tag{A8}$$

The approximate account for aerosol absorption effects is performed multiplying $R_{ss}$ (see Equation (A2)) by the single scattering albedo $\omega_0$ [17]. The accuracy of Equations (A1)–(A3) can be further improved using the truncation approximation as discussed in [16].

The transmission function $T(\mu_0, \mu)$ is approximated as follows:

$$T(\mu_0, \mu) = t^m, \tag{A9}$$

where $t$ is calculated using the following approximation [16]:

$$t = e^{-B\tau}. \tag{A10}$$

Here,

$$B = \frac{1}{2}\int_{\frac{\pi}{2}}^\pi p(\theta)\sin\theta\cos\theta d\theta \tag{A11}$$

is the so-called backscattering fraction. The atmospheric spherical albedo $r_a$ is found using the approximation proposed in [11]:

$$r_a = \left(Me^{-\frac{\tau}{\varsigma}} + Ne^{-\frac{\tau}{\kappa}} + D\right)\tau. \tag{A12}$$

The coefficients of polynomial expansions of all coefficients ($M, N, D, \varsigma, \kappa$) in Equation (A12) with respect to the value of $g$ are given in [11].

One can see that the reflection function depends on the atmospheric optical thickness, which can be presented in the following form:

$$\tau(\lambda) = \tau_{mol}(\lambda) + \tau_{aer}(\lambda). \tag{A13}$$

The molecular optical thickness can be approximated as [18,19]:

$$\tau_m(\lambda) = q\lambda^{-\upsilon} \tag{A14}$$

at the normal pressure $p_0$ and temperature $T_0$. Here, $q = 0.008735$, $v = 4.08$, and the wavelength is in microns. We derive the value of molecular optical thickness at another pressure level $p$ using the following expression: $\tau_{mol}(\lambda) = \hat{p}\tau_m(\lambda)$, where $\hat{p} = \frac{p}{p_0}$, $p$ is the site pressure, $p_0 = 1013.25$ mb. The site pressure is calculated using the following equation: $p = p_0 \exp\left(-\frac{z}{H}\right)$. Here, $z$ is the height of the underlying surface provided in OLCI files, and $H = 7.64$ km is the scale height.

It follows for the aerosol optical thickness (AOT) [4]:

$$\tau_{aer}(\lambda) = \beta\left(\frac{\lambda}{\lambda_0}\right)^{-\alpha}, \tag{A15}$$

where $\lambda_0 = 0.5$ μm, and the pair $(\alpha, \beta)$ represents the Angström parameters. We do not make an attempt to derive the pair $(\alpha, \beta)$ over snow. These values must be assumed ahead of retrievals (e.g., using aerosol climatology [20], ground measurements, or aerosol forecasts). The statistical results for the values $\alpha, \beta$ over various Greenland AERONET [21] stations are given in Figures A1 and A2. It follows that, over Greenland, the value of $\beta = \tau_{aer}(\lambda_0)$ is in the range 0.02–0.12 on average (see Figure A1). For our BBA albedo retrievals reported in this paper, we assume that $\beta = 0.07$ independently on location and time. The AERONET monthly statistics show that $\alpha$ is in the range 1.0–1.6 over Greenland (see Figure A2). Therefore, we assume the value of $\alpha = 1.3$ in our retrievals.

The phase function can be presented in the following form:

$$p(\theta) = \frac{\tau_{mol}p_{mol}(\theta) + \tau_{aer}p_{aer}(\theta)}{\tau_{mol} + \tau_{aer}}, \tag{A16}$$

where

$$p_{mol}(\theta) = \frac{3}{4}\left(1 + \cos^2\theta\right) \tag{A17}$$

is the molecular scattering phase function, and $p_{aer}(\theta)$ is the aerosol phase function. We represent this function as:

$$p_{aer} = \frac{1 - g_{aer}^2}{\left(1 - 2g_{aer}\cos\theta + g_{aer}^2\right)^{\frac{3}{2}}}. \tag{A18}$$

Therefore, it follows for the asymmetry parameter:

$$g = \frac{\tau_{aer}}{\tau_{mol} + \tau_{aer}}g_{aer}. \tag{A19}$$

The parameter $g_{aer}$ varies with location, time, aerosol, type, etc. We assume that it can be approximated by the following equation:

$$g_{aer} = g_0 + g_1 e^{-\frac{\lambda}{\lambda_0}}. \tag{A20}$$

The coefficients in this equation (as derived from multiple year AERONET observations over Greenland, see Figure A3) are as follows:

$$g_0 = 0.5263, \ g_1 = 0.4627, \ \lambda_0 = 0.4685 \ \text{μm}. \tag{A21}$$

The parameter $B$ for the Henyey-Greenstein can be calculated analytically [see Equation (A11)]:

$$B(g) = \frac{1 - g}{2g}\left(\frac{1 + g}{\sqrt{1 + g^2}} - 1\right). \tag{A22}$$

It follows from Equation (A22) that that B(1) = 0, as it should be. Also, we have from Equation (A22) at g → 0:

$$B(g) = \frac{1}{2} + \frac{g(g^2 - 3)}{2(1 + g^2 + (1 - g^2)\sqrt{1 + g^2})}. \tag{A23}$$

Therefore, it follows that B (0) = 0.5, as it should be for the symmetric phase functions.

The system of equations given above enables the calculation of underlying snow-atmosphere reflectance as a function of the aerosol optical thickness for a known value of the snow spherical albedo (see Equation (9)).

As far as gaseous transmission is concerned, we propose to use the following exponential approximation [22]:

$$T_{ozone} = \exp(-m\gamma\tau_{ozone}), \tag{A24}$$

where,

$$\gamma = \frac{c_{O3}}{c}. \tag{A25}$$

Here, $c_{O3}$ is the ozone concentration provided in the OLCI satellite file (with account for units), and $\tau_{ozone}$ is the vertical optical depth of ozone at the concentration $c = 405$ DU. In particular, to transfer from OLCI O3 units (kg/m$^2$) to Dobson units (DU), we multiply OLCI O3 concentration by a constant factor equal to $4.6729 \times 10^4$. Therefore, the total ozone load 300 DU corresponds to $6.42 \times 10^{-3}$ kg/m$^2$. The values $\tau_{ozone}$ calculated for all OLCI channels at $c = 405$ DU with account for the instrument response function are given in Table A1.

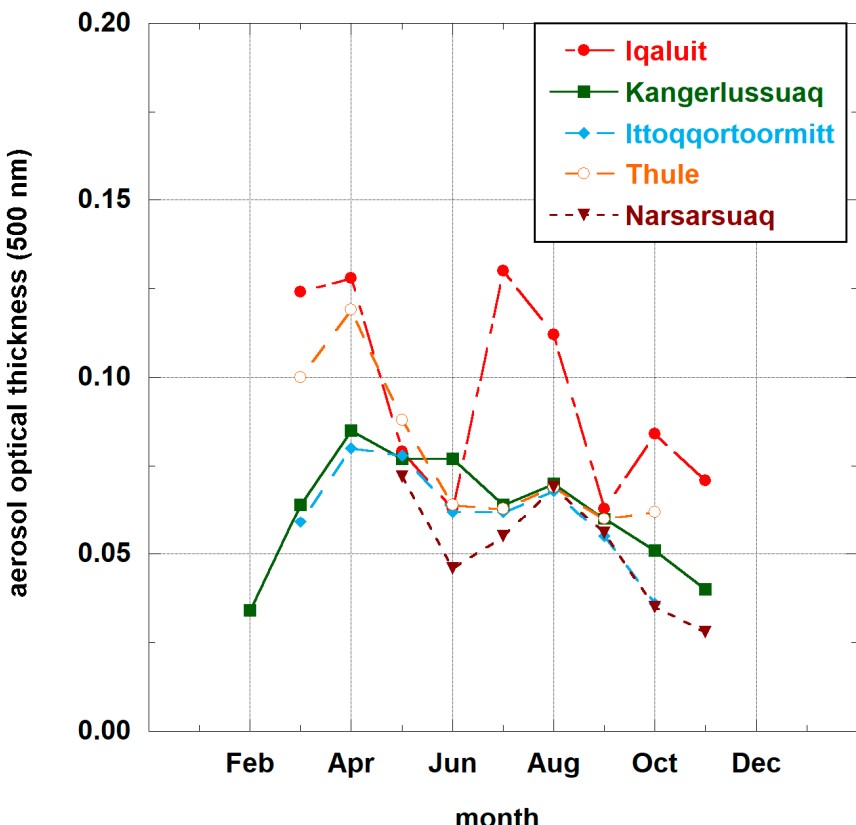

**Figure A1.** The statistical properties of the aerosol optical thickness at 500 nm over various AERONET stations in Greenland. The results are derived from Level 2 (Verison 3 [22] AERONET data) for years 2007–2017.

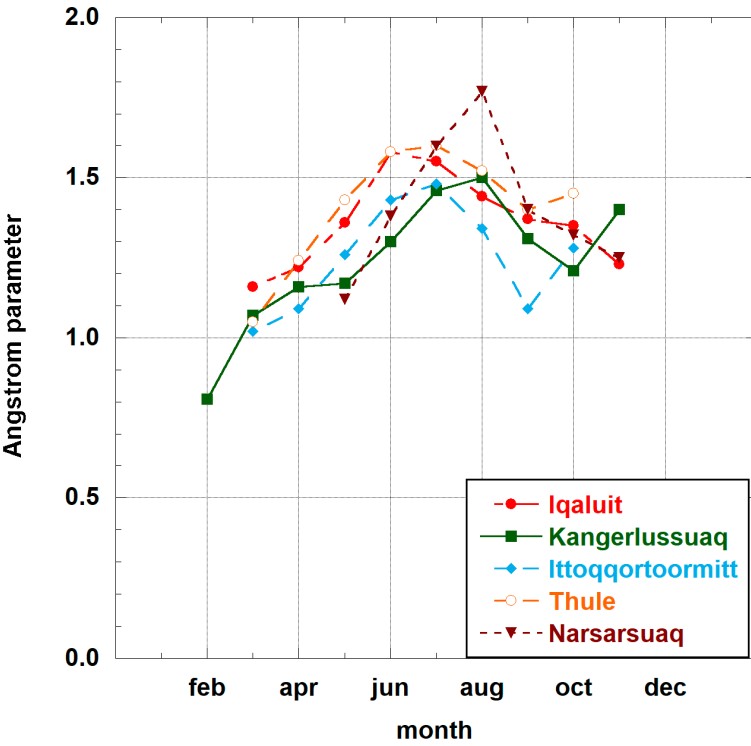

**Figure A2.** The statistical properties of the Angstroem parameter for atmospheric aerosol over various AERONET stations in Greenland. The results are derived from Level 2 (Verison 3 [22] AERONET data) for years 2007–2017.

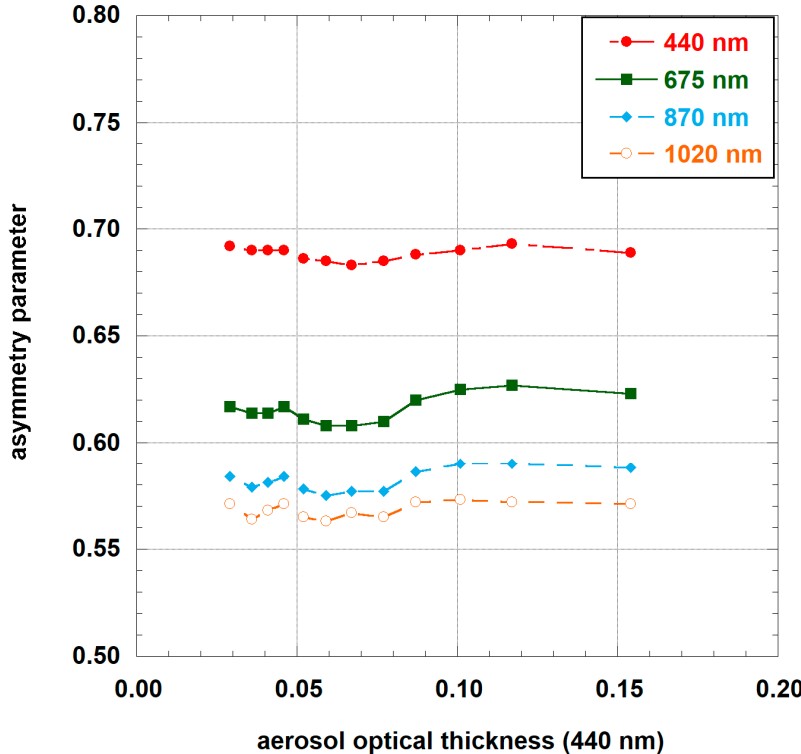

**Figure A3.** The asymmetry parameter climatology (2007–2017) for five considered sites in Greenland (version 3 [22,23]) data, Level 1.5 retrievals with the residual error < 5%, and AOT (440 nm) < 0.20). Total number of retrievals is 5316 divided in 12 groups with 443 asymmetry parameters in each group.

**Table A1.** The spectral dependence of ozone vertical optical thickness $\tau_{ozone}$ in terrestrial atmosphere at the ozone load equal to 405 Dobson units (DU). The results are derived assuming particular shapes of temperature, pressure, and ozone concentration vertical distribution as discussed in [24]. We find that the variation of profiles does not change the value of $\tau_{ozone}$ significantly.

| $\lambda$, nm | $\tau_{ozone}$ |
|---|---|
| 400.00000 | $1.378170469 \times 10^{-4}$ |
| 412.50000 | $3.048780958 \times 10^{-4}$ |
| 442.50000 | $1.645714060 \times 10^{-3}$ |
| 490.00000 | $8.935947110 \times 10^{-3}$ |
| 510.00000 | $1.750535146 \times 10^{-2}$ |
| 560.00000 | $4.347104369 \times 10^{-2}$ |
| 620.00000 | $4.487130794 \times 10^{-2}$ |
| 665.00000 | $2.101591797 \times 10^{-2}$ |
| 673.75000 | $1.716230955 \times 10^{-2}$ |
| 681.25000 | $1.466298300 \times 10^{-2}$ |
| 708.75000 | $7.983028470 \times 10^{-3}$ |
| 753.75000 | $3.879744653 \times 10^{-3}$ |
| 761.25000 | $2.923775641 \times 10^{-3}$ |
| 764.37500 | $2.792211429 \times 10^{-3}$ |
| 767.50000 | $2.729651478 \times 10^{-3}$ |
| 778.75000 | $3.255969698 \times 10^{-3}$ |
| 865.00000 | $8.956858078 \times 10^{-4}$ |
| 885.00000 | $5.188799343 \times 10^{-4}$ |
| 900.00000 | $6.715773241 \times 10^{-4}$ |
| 940.00000 | $3.127781417 \times 10^{-4}$ |
| 1020.00000 | $1.408798425 \times 10^{-5}$ |

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
