# Peer review of "The Determination of Snow Albedo from Satellite Measurements Using Fast Atmospheric Correction Technique"

_remotesensing, doi:10.3390/rs12020234_

Round 1

Reviewer 1 Report

 This work is an improvement of a previous one. The improvement consisting of a new method for atmospheric correction of Sentinel 3 data, applied to surfaces covered by polluted snow. The improvement consists considering the role of aerosols, which was neglected in the previous version.

The work has a sound theoretical basis. However, the theory is sometimes difficult to follow. Moreover, the results obtained do not seem to be as good as the authors claim.

Before accepting the paper for publication, I think the authors should revise the manuscript according to the following points:

Regarding the theoretical model:

According to the authors (lines 231-233): “Here, we use the improved atmospheric correction, which explicitly accounts for molecular/aerosol light scattering and absorption effects”. In the previous version (reference 6), the estimation of the albedo is based on the calculation RBOA for Sentinel 3 (OLCI L1b product). Which effects are not taken into account in the calculation of RBOA in the previous version: aerosols, molecular scattering,…? Could the authors be more specific? This is important in order to fully understand the improved atmospheric correction. When calculating the albedo, the authors state that the previous atmospheric correction over clean snow is valid. In the Discussion (on line 333) the authors state that Rs in Equation (2) is TOA reflectance. However, in reference 6, the authors use OLCI L1b as input to calculate the albedo for clean snow. Is Rs in Equation (2) TOA reflectance or BOA reflectance? Why can OLCI L1b be used in the case of clean snow and not in the case of polluted snow? The authors must explain the physical reasons. Why is rs in Equation (7) replaced by Rs in the numerator only to obtain Equation (8)? Line 142: why cannot R0 be calculated like this in the case of clean snow? Line 172. When calculating the incident solar flux, the authors use Equation (12). Why do the authors use Equation (12) and not the F directly obtained from SBDISORT like they apparently do in the case of clean snow? A motivation must be given for the interpolation equations used to calculate the spherical albedo (Equation (13) and Equation (14)). Lines 197-198: why are the coefficients of Equation (14) calculated in a differente way depending of the value of Rmeasured (1020 nm). What is the reason for the threshold value of Rmesured (1020nm) < 0.5?

Regarding the validation results:

On Lines 237-238 the authors state: “As it follows from Fig.1b, the variation of AOT (500nm) in the range 0.07-0.35 does not change the plane albedo retrieval accuracy”. I do not understand this statement. In Fig. 1 b the accuracy does change with AOT. On line 242 the authors state that the value of AOT for the case of Fig. 1 is 0.125. It is remarkable that precisely for this value, the results of the new atmospheric correction do not seem to be much better than the older one and the difference with in-situ albedo is the biggest. Why? Regarding Figure 3: the data includes clean snow and polluted snow, I guess. Do the high albedo values correspond to clean snow? If so, I guess that high albedo values were calculated using the clean snow procedure. This has to be explained. Regarding Figure 3: It is noteworthy that the discrepancy between in-situ data and SICE is greater for lower albedo values (polluted ice). Could you please comment on this? On line 308, the authors compare the results with those from MOD10A1. It has recently been pointed out that MOD10A1 data exhibit extremely low unrealistic values the effect of which can be masked using a maximum filter (Sensors 2019, 19(16), 3569; https://doi.org/10.3390/s19163569). Figure 4 shows some MOD10A1 extremely low values. This should be commented on.

Minor remarks:

Line 99: edit “coefficient known”. The authors should provide the values of bulk ice absorption coefficient for the wavelengths or provide the reference. On Lines 115-116, the authors state: “the plane albedo is the integral of the plane albedo .... “. I am afraid you mean: the spherical albedo is defined via the integral of the plane albedo with respect to the solar zenith angle. Am I right?

Reviewer 2 Report

The paper introduced an improved snow albedo retrieval algorithm by including an atmospheric correction method.The significance of the topic is clear and the theory is advanced; however , the description of the theory can be clearer and the validation analysis could be more completed. Iwould recommend major revisi on before considering for publication.

1. The symbols in the equations have not been clearly explained.Although some of them can be found in reference [6],I still recommend providing at able to organize and state the meaning of the symbols.

2.The description of Fig1.shows some conflict in atitude.The authors states the results from the improved method has a higher accuracy. In fact, the S3Snow and 0. 125 retrievals donot have much difference, just as the authors claimed that the results from the variation of AOT do not change accuracy. This validation case seems not sufficient to prove the advancement of the improved method in spectral albedo practice.

2. Fig3.Not only the marker color, but also the marker type is suggested in legend.

3.Fig3.The authors claimed a higher accuracy than MOD10A1.However, there is not any background information provided in previous sections about the MOD10A1 data, including the algorithm and match-up criterion.

Besides, the SICE demonstrates more outliers than MOD10A1, analysis is recommended about this.

4.FromTable3, several sites, including QAS_U, NUK_L, UPE_L, UPE_U, show very poor validation results. I have not seen the scatterplots comparison for these sites or explanations.

5.Fig4 and 5, do the comparison with different products contain the same sample size?

6.line302,'station frame correction' was used to explain the bias between the retrieval and the groundmeasurements. However, not much detailed information was provided about the station frame correction. Iwould suggest adding more background information here.

7.The method contains many simpli fication and empirical coefficients.I recommend the authors add more analysis about the limitations of the method in spatial extent.

Reviewer 3 Report

Overall this paper presents an incremental improvement to a satellite snow albedo retreival algorithm and is a clear, well- written manuscript.  The reason I downgraded the novelty and significance to "average" is because this is only an incremtnal improvement.  However, it is my opinion this paper should be published in Remote Sensing with very minor changes.

Suggestions for improvement:

Introduction, first sentence (line 41):  The incendiary phrase "global warming" is used. Climate change is not forcing warming everywhere on the planet--it is forcing warming primarily at high altitudes and latitudes (exactly the areas used for validation in this paper) due to lower water vapor content in these levels and regions.  Suggest "global warming" be changed to "climate change" or "warming at high latitudes and altitudes."

Line 99:  The word "coefficient" has an unecessary space between "co" and "efficient."

The paragraph that cites Shettle Fenn [13], line 178:  I am absolutely baffled why this paper would use the rural aerosol model from Shettle Fenn to characterize Arctic aerosols.  The OPAC/GADS characterization at least contains more updated optical properties for both Artic and Antarctic aerosols.  Suggest those be used for the SBDART calculations.  Citations below.

Hess, M., P. Koepke, and I. Schult, 1998: Optical Properties of Aerosols and Clouds: The software package OPAC. Bull.Amer. Meteor. Soc., 79, 831–844.

Koepke, P., M. Hess, I. Schult, and E. P. Shettle, 1997: Global Aerosol Data Set. Max-Planck-Institut f€ur Meteorologie Rep. 243, 44 pp. [Available online at http://rascin.net/sites/default/files/downloads/MPI-Report_243.pdf.]

Round 2

Reviewer 1 Report

The authors have reviewed the paper taking into account the comments of the referee.

Two comments:

1) One question remains unanswered:

I would like the authors to provide the values of the absorption coefficients of snow for the wavelenghts used or provide an adequate reference. Reference 6 is not appropriate. I have not been able to find the values of the coeff. of absorption of ice in that reference.

2) one minor remark:

Line 383: I am afraid you shoud replace "Than" by "Then".

Author Response

Please, see attached

Reviewer 2 Report

Thank the authors for the clarifications and modifications in the revised version. I would recommend for the publication of the manuscript in current form.

Author Response

Thanks for your support.